The clinical significance of collagen family gene expression in esophageal squamous cell carcinoma

Li Jieling 1
Wang Xiao 2
Zheng Kai 1
Liu Ying 2
Li Junjun 1
Wang Shaoqi 3
Liu Kaisheng 2
Song Xun 1
Li Nan 2
Xie Shouxia 2 szshouxia@163.com
Wang Shaoxiang 1 wsx@szu.edu.cn
1 School of Pharmaceutical Sciences, Shenzhen University Health Science Center , Shenzhen , China
2 Department of Pharmacy, The Second Clinical Medical College (Shenzhen People’s Hospital), Jinan University , Shenzhen , China
3 Department of Oncology, Hubei Provincial Corps Hospital, Chinese People Armed Police Forces , Wuhan , China
Kumar Abhishek
Electronic publication date: 2019 Oct 4
Publication date: 2019
Volume: 7
Electronic Location ID: e7705
Received 2019 Mar 20; Accepted 2019 Aug 19
Copyright: © 2019 Li et al.
Copyright year: 2019
Copyright holder: Li et al.
License: This is an open access article distributed under the terms of the Creative Commons Attribution License, which permits unrestricted use, distribution, reproduction and adaptation in any medium and for any purpose provided that it is properly attributed. For attribution, the original author(s), title, publication source (PeerJ) and either DOI or URL of the article must be cited.
License URL: https://creativecommons.org/licenses/by/4.0/

Keywords: ESCC, TCGA, GEO, Gene expression, Overall survival, Collagen

Funding: National Natural Science Foundation of China 81602625 Natural Science Foundation of Guangdong Province 2016A030310096, 2018A030313122 Science and Technology Planning Project of Guangdong Province 2017A010105013 Pearl River S&T Nova Program of Guangzhou 201710010011 Shenzhen Science and Technology Project JCYJ20170302145059926, JCYJ20180305163658916, JCYJ20180228175059744 This work was financially supported by the National Natural Science Foundation of China (Nos. 81602625), the Natural Science Foundation of Guangdong Province (2016A030310096, 2018A030313122), the Science and Technology Planning Project of Guangdong Province (2017A010105013), the Pearl River S&T Nova Program of Guangzhou (201710010011), and the Shenzhen Science and Technology Project (JCYJ20170302145059926, JCYJ20180305163658916, JCYJ20180228175059744). The funders had no role in study design, data collection and analysis, decision to publish, or preparation of the manuscript.

==============================
Background

Esophageal squamous cell carcinoma (ESCC) is a subtype of esophageal cancer with high incidence and mortality. Due to the poor 5-year survival rates of patients with ESCC, exploring novel diagnostic markers for early ESCC is emergent. Collagen, the abundant constituent of extracellular matrix, plays a critical role in tumor growth and epithelial-mesenchymal transition. However, the clinical significance of collagen genes in ESCC has been rarely studied. In this work, we systematically analyzed the gene expression of whole collagen family in ESCC, aiming to search for ideal biomarkers.

Methods

Clinical data and gene expression profiles of ESCC patients were collected from The Cancer Genome Atlas and the gene expression omnibus databases. Bioinformatics methods, including differential expression analysis, survival analysis, gene sets enrichment analysis (GSEA) and co-expression network analysis, were performed to investigate the correlation between the expression patterns of 44 collagen family genes and the development of ESCC.

Results

A total of 22 genes of collagen family were identified as differentially expressed genes in both the two datasets. Among them, COL1A1, COL10A1 and COL11A1 were particularly up-regulated in ESCC tissues compared to normal controls, while COL4A4, COL6A5 and COL14A1 were notably down-regulated. Besides, patients with low COL6A5 expression or high COL18A1 expression showed poor survival. In addition, a 7-gene prediction model was established based on collagen gene expression to predict patient survival, which had better predictive accuracy than the tumor-node-metastasis staging based model. Finally, GSEA results suggested that collagen genes might be tightly associated with PI3K/Akt/mTOR pathway, p53 pathway, apoptosis, cell cycle, etc.

Conclusion

Several collagen genes could be potential diagnostic and prognostic biomarkers for ESCC. Moreover, a novel 7-gene prediction model is probably useful for predicting survival outcomes of ESCC patients. These findings may facilitate early detection of ESCC and help improves prognosis of the patients.

Introduction

Esophageal cancer is the seventh most commonly diagnosed cancer and the sixth leading cause of cancer death (Bray et al., 2018). It is classified into two histological subtypes, esophageal adenocarcinoma and esophageal squamous cell carcinoma (ESCC), the latter of which is the predominant type worldwide (Pennathur et al., 2013). Despite the effective treatments (e.g., surgery, chemotherapy and radiotherapy) for ESCC, the 5-year survival rates of patients with advanced ESCC are still less than 20% (Codipilly et al., 2018). However, the survival rates could be improved to over 80% if patients were diagnosed with an early stage (Lao-Sirieix & Fitzgerald, 2012; Wang et al., 2004). Although a few tumor markers, carcinoembryonic antigen, carbohydrate antigen 19-9, and squamous cell carcinoma antigen, have been used in the diagnosis of ESCC, they are not suitable for early detection due to the lack of sensitivity (Kosugi et al., 2004). Thus, it is urgent to search for novel biomarkers to help early detection of ESCC and improve survival rates of the patients.

Collagen is the most abundant extracellular matrix protein that promotes cell growth and provides mechanical resilience of connective tissues (Sorushanova et al., 2018). The collagen family comprises 28 types with different α chains encoded by more than 40 genes (Ricard-Blum, 2011). It has been reported that the expression of collagen-encoding genes was significantly related to the prognosis of certain types of cancers (Giussani et al., 2018; Liu et al., 2018; Rong et al., 2018; Shen et al., 2016; Zhang et al., 2018c). In addition, a couple of collagen genes, such as COL11A1 and COL6A1, were expressed aberrantly in ESCC tissues and possibly affected the progression of ESCC (Fan et al., 2012; He et al., 2017; Zhang et al., 2018a). However, most of these works focused on specific collagen genes, and the potential roles of other members remain to be clarified.

Here, we provided a systematic analysis of gene expression of the whole collagen family and its corresponding clinical significance in ESCC. Clinical data and gene expression profiles of ESCC patients were extracted from The Cancer Genome Atlas (TCGA) and the gene expression omnibus (GEO), two public databases with substantial information about cancers. Different bioinformatics methods, including differential expression analysis, survival analysis, pathway analysis and co-expression network analysis were used to analyze the data to sift important hits possibly involved in the initiation and development of ESCC. According to collagen family genes, we also established a prediction model with high performance to predict the prognosis of ESCC patients. Collectively, our works mainly explored the relation of collagen gene expression to ESCC and illuminated the potential mechanism.

Materials and Methods

Patient data

Basic data of ESCC patients were downloaded from the TCGA database (https://portal.gdc.cancer.gov/) and the GSE53625 dataset of the GEO database; 95 cases from TCGA and 179 cases from GSE53625. Univariate and multivariate Cox regression analyses were carried out to investigate the correlation between overall survival and clinicopathological characteristics of the patients by SPSS (v23.0). The relations between collagen family gene expression and clinicopathological characteristics of the patients were examined using Pearson correlation analysis via SPSS.

Differential expression analysis

Gene expression profiles of tumor and adjacent normal tissues in ESCC patients were also obtained from the two datasets. 81 of 95 patient cases in TCGA and all patient cases in GEO had RNA-sequence data. In total, 81 tumor samples with 11 normal controls from TCGA and 179 tumor samples with 179 normal controls from GEO (Li et al., 2014) were included in analysis (each sample was taken from a different patient). Differential expression analysis was conducted using the edgeR (Robinson, McCarthy & Smyth, 2010) and the limma (Ritchie et al., 2015) packages, respectively, for TCGA and GEO data by R software (R Core Team, 2018). Gene expression levels were normalized by the calcNormFactors function in edgeR (Law et al., 2016) and by the normalizeBetweenArrays function in limma (Smyth & Speed, 2003), to make sure the expression distributions of each sample are similar across the entire matrix. Then based on the exact test in edgeR which is analogous to Fisher’s exact test (Robinson, McCarthy & Smyth, 2010) and the Empirical Bayes statistical test in limma (Phipson et al., 2016), fold change, P-value and false discovery rate (FDR) (or adjusted P-value) were figured out to show the expression difference between tumor and normal samples. Genes with P < 0.05 and FDR < 0.05 were considered as differentially expressed genes (DEGs). Accordingly, DEGs of collagen family were identified. Then heatmaps, boxplots and Venn diagram were drawn by R software.

Survival analysis

First, hazard ratio (HR) and P-value of each DEG of collagen family were figured out based on gene expression and overall survival of patients by the univariate Cox regression model with the survival package through R software. The HR is an estimate of the ratio of the hazard rate in the treated versus the control group (Spruance et al., 2004), while in this study it is defined as the hazard in the high expression group divided by the hazard in the low expression group. HR > 1 and HR < 1 mean higher expression of the gene is associated with worse and better overall survival, respectively. Survival curves were plotted according to the Kaplan–Meier method and compared by the log-rank test using the survival and the q-value packages in R. P < 0.05 was considered statistically significant.

Prediction models

Prediction models were established to predict patient survival based on gene expression of 22 DEGs of collagen family and overall survival of patients by the multivariate Cox regress analysis with the survival package via R software. Several candidate genes were eventually selected out by the analysis to form the model, with a formula calculating the risk score of each patient. The general formula is given below: (1) Riskscore=∑i=1n⁡Coefi×Expi

where n, Coef and Exp indicate the number of included genes, the coefficient of each gene, and gene expression level, respectively. The coefficients were estimated based on the relative contributions of each collagen gene. A patient’s risk score was calculated as the sum of the expression levels of each gene multiplied by its corresponding coefficient. Similar methods have been adopted by earlier studies (Beer et al., 2002; Lossos et al., 2004; Wang et al., 2018). Then, receiver operating characteristic (ROC) curves were plotted based on the risk scores and overall survival of patients by the survivalROC package in R, with area under curve (AUC) values which represented the accuracy of predicting 3-year survival. Also, survival curves were obtained by dividing the patients into high- and low-risk groups according to the median risk score using the survival package.

Pathway analysis

Potential mechanism of collagen family genes was explored by the gene sets enrichment analysis (GSEA), a method to determine whether members of a previously defined gene set are correlated with the phenotypic class distinction (Subramanian et al., 2005). GSEA was conducted using the gene expression profiles of patients’ tumor samples via javaGSEA software (http://software.broadinstitute.org/gsea/downloads.jsp), and the patient samples were divided into high- and low-risk groups in half according to the risk scores obtained by the collagen-DEGs-based prediction models (Chai et al., 2018; Zhang et al., 2017; Zhao et al., 2017). Oncogenic Signatures Gene Sets (v6.2), Hallmark Gene Sets (v6.2) and KEGG Gene Sets (v6.2) (http://software.broadinstitute.org/gsea/msigdb/collections.jsp) were, respectively, used as references. Based on these gene sets databases, the expression profiles were analyzed to find out if a set of genes were mostly up-regulated (or down-regulated) in the high-risk group (or low-risk group). Normalized enrichment score reflected the degree to which a gene set was overrepresented in the groups, and gene sets in the results with P < 0.05 and FDR < 0.25 were considered as significant ones (Subramanian et al., 2005).

Co-expression network analysis

Patients’ tumor samples from TCGA were separated into high- and low-risk groups by the risk scores calculated by the 7-gene prediction model. Risk-score-based DEGs that were differentially expressed between the two groups were determined using the gene expression profiles of tumor samples by the same method as differential expression analysis. Then the relationships between collagen family genes and the risk-score-based DEGs as well as the representative enriched gene sets from GSEA were assessed by the Weighted Gene Co-Expression Network Analysis (WCGNA) with the WGCNA package through R software, which is a method to describe the correlation patterns among genes across different samples (Langfelder & Horvath, 2008). Genes of each gene set were extracted from http://software.broadinstitute.org/gsea/msigdb/genesets.jsp. Finally, the genes co-expressed with collagen family genes were obtained, and the networks of them were drawn via Cytoscape (http://www.cytoscape.org/, v3.7.1).

Results

Clinicopathological information of the ESCC patients

A total of 95 patient cases in TCGA and 179 cases in GEO were collected and analyzed by univariate and multivariate Cox regression analyses. As a result, poor overall survival was significantly correlated with sex, tumor-node-metastasis (TNM) stage and N stage in TCGA (P = 0.020, P = 0.015, and P = 0.012, respectively) (Table 1), and was notably associated with age, TNM stage and N stage in GEO (P = 0.021, P < 0.001, and P = 0.030, respectively) (Table 2). Besides, investigation into the correlation between collagen family gene expression and the clinicopathological characteristics revealed that the expression of several collagen genes was significantly related to advanced TNM stages or tumor grades. (Tables 3 and 4).

Table 1 Univariate and multivariate analyses of clinicopathological characteristics for overall survival in ESCC patients from the TCGA dataset (N = 95).

Variables	n (%)	Univariate analysis	Multivariate analysis	
HR (95% CI)	P	HR (95% CI)	P	
Age						
 <60	56 (58.9%)	1 (Reference)				
 ≥60	39 (41.1%)	1.296 [0.631–2.662]	0.461			
Sex						
 Male	80 (84.2%)	1 (Reference)		1 (Reference)		
 Female	15 (15.8%)	0.175 [0.041–0.756]	0.020	0.206 [0.043–0.978]	0.047	
TNM Stage						
 I + II	63 (66.3%)	1 (Reference)		1 (Reference)		
 III + IV	31 (32.6%)	2.443 [1.191–5.011]	0.015	0.921 [0.321–2.643]	0.879	
 Missing	1 (1.1%)					
T Stage						
 T1 + T2	40 (42.1%)	1 (Reference)				
 T3 + T4	54 (56.8%)	1.351 [0.649–2.811]	0.422			
 Missing	1 (1.1%)					
Tumor grade						
 G1 + G2	65 (68.4%)	1 (Reference)				
 G3	21 (22.1%)	0.736 [0.277–1.950]	0.537			
 Missing	9 (9.5%)					
N Stage						
 N0 + N1	84 (88.4%)	1 (Reference)		1 (Reference)		
 N2 + N3	9 (9.5%)	3.265 [1.302–8.189]	0.012	6.738 [1.493–30.399]	0.013	
 Missing	2 (2.1%)					
Tumor location						
 Upper + Middle	50 (52.6%)	1 (Reference)				
 Lower	44 (46.3%)	0.958 [0.448–2.051]	0.913			
 Missing	1 (1.1%)					
Alcohol use						
 No	25 (26.3%)	1 (Reference)		1 (Reference)		
 Yes	68 (71.6%)	2.172 [0.751–6.276]	0.152	4.755 [1.054–21.457]	0.043	
 Missing	2 (2.1%)					
Tobacco use						
 No	44 (46.3%)	1 (Reference)		1 (Reference)		
 Yes	51 (53.7%)	1.965 [0.901–4.285]	0.089	1.095 [0.440–2.725)	0.845	
Race						
 Asian	45 (47.4%)	1 (Reference)		1 (Reference)		
 White + Other	47 (49.5%)	1.570 [0.688–3.581]	0.284	2.021 [0.782–5.223]	0.146	
 Missing	3 (3.2%)					
Notes:

Characteristics with P < 0.3 in the univariate analysis were further screened in the multivariate analysis.

HR, hazard ratio; CI, confidence interval; TNM stage, tumor-node-metastasis stage; T stage, stage of tumor invasion; N stage, stage of regional lymph node invasion.

Table 2 Univariate and multivariate analyses of clinicopathological characteristics for overall survival in ESCC patients from the GEO dataset (N = 179).

Variables	n (%)	Univariate analysis	Multivariate analysis	
HR (95% CI)	P	HR (95% CI)	P	
Age						
 <60	91 (50.8%)	1 (Reference)		1 (Reference)		
 ≥60	88 (49.2%)	1.574 [1.072–2.311]	0.021	1.451 [0.980–2.147]	0.063	
Sex						
 Male	146 (81.6%)	1 (Reference)				
 Female	33 (18.4%)	1.277 [0.798–2.044]	0.307			
TNM Stage						
 I + II	87 (48.6%)	1 (Reference)		1 (Reference)		
 III + IV	92 (51.4%)	2.155 [1.448–3.207]	<0.001	2.066 [1.322–3.228]	0.001	
T Stage						
 T1 + T2	39 (21.8%)	1 (Reference)				
 T3 + T4	140 (78.2%)	1.091 [0.687–1.732]	0.712			
Tumor grade						
 G1 + G2	99 (55.3%)	1 (Reference)		1 (Reference)		
 G3	80 (44.7%)	1.391 [0.951–2.037]	0.089	1.269 [0.860–1.873]	0.230	
N Stage						
 N0 + N1	145 (81.0%)	1 (Reference)		1 (Reference)		
 N2 + N3	34 (19.0%)	1.644 [1.048–2.577]	0.030	1.062 [0.644–1.751]	0.814	
Tumor location						
 Upper + Middle	117 (65.4%)	1 (Reference)				
 Lower	62 (34.6%)	0.823 [0.546–1.242]	0.354			
Alcohol use						
 No	73 (40.8%)	1 (Reference)				
 Yes	106 (59.2%)	0.864 [0.588–1.269]	0.456			
Tobacco use						
 No	65 (36.3%)	1 (Reference)		1 (Reference)		
 Yes	114 (63.7%)	0.749 [0.508–1.105]	0.145	0.753 [0.505–1.122]	0.163	
Pneumonia						
 No	164 (91.6%)	1 (Reference)				
 Yes	15 (8.4%)	1.425 [0.719–2.824]	0.310			
Notes:

Characteristics with P < 0.3 in the univariate analysis were further screened in the multivariate analysis.

HR, hazard ratio; CI, confidence interval; TNM stage, tumor-node-metastasis stage; T stage, stage of tumor invasion; N stage, stage of regional lymph node invasion.

Table 3 Correlation of collagen family gene expression and clinicopathological characteristics of ESCC patients from the TCGA dataset.

Gene	Ag ≥ 60	Sex (Female)	TNM stage III/IV	N stage (N1 + N2)	Tumor grade (G3)	Tumor location (Lower)	
COL1A1		−0.222*0.048					
COL1A2		−0.222*0.048					
COL2A1							
COL3A1		−2.225*0.045					
COL4A1							
COL4A2							
COL4A3							
COL4A4							
COL4A5							
COL4A6							
COL5A1							
COL5A2		−0.231*0.039					
COL5A3		−0.229*0.041					
COL6A1							
COL6A2							
COL6A3							
COL6A5							
COL6A6							
COL7A1					−0.226*0.046	−0.226*0.046	
COL8A1							
COL8A2							
COL9A1							
COL9A2							
COL9A3		0.318**0.004					
COL10A1							
COL11A1							
COL11A2							
COL12A1						−0.288*0.010	
COL13A1							
COL14A1							
COL15A1							
COL16A1			−0.280*0.013		−0.280*0.013		
COL17A1			−0.299**0.008		−0.299**0.008		
COL18A1							
COL19A1				0.367**0.00			
COL20A1							
COL21A1		0.243*0.030					
COL22A1							
COL23A1							
COL24A1							
COL25A1							
COL26A1							
COL27A1	−0.245*0.02						
COL28A1							
Notes:

Superscripts of the correlation coefficients represent P-values.

* Correlation with P < 0.05.

** Correlation with P < 0.01.

Table 4 Correlation of collagen family gene expression and clinicopathological characteristics of ESCC patients in GEO.

Gene	Age ≥ 60	Sex (Female)	TNM stage III + IV	N stage (N1 + N2)	Tumor grade (G3)	Tumor location (Lower)	
COL1A1							
COL1A2							
COL2A1							
COL3A1							
COL4A1							
COL4A2							
COL4A3					0.149*0.046	−0.162*0.030	
COL4A4						−0.168*0.024	
COL4A5							
COL4A6							
COL5A1							
COL5A2							
COL5A3						0.167*0.026	
COL6A1							
COL6A2							
COL6A3							
COL6A5					−0.173*0.020		
COL6A6							
COL7A1							
COL8A1		0.188*0.012					
COL8A2							
COL9A1							
COL9A2				−0.175*0.019			
COL9A3					0.162*0.030		
COL10A1					−0.151*0.044		
COL11A1							
COL11A2							
COL12A1							
COL13A1							
COL14A1							
COL15A1							
COL16A1							
COL17A1							
COL18A1							
COL19A1					0.174*0.020		
COL20A1							
COL21A1			−0.163*0.029				
COL22A1							
COL23A1							
COL24A1							
COL25A1						0.147*0.049	
COL26A1	0.174*0.020					0.206**0.006	
COL27A1		−0.174*0.020					
COL28A1							
Notes:

Superscripts of the correlation coefficients represent P-values.

* Correlation with P < 0.05.

** Correlation with P < 0.01.

Identification of DEGs of collagen family in ESCC tissues

Differential expression analysis showed that more than 2/3 of the 44 collagen family genes were up-regulated in tumor tissues in both TCGA and GEO (Tables S1 and S2). A total of 22 members in TCGA and 35 members in GEO were identified as DEGs, and their expression patterns were shown by heatmaps (Figs. 1A and 1B). Then the Venn diagram demonstrated that there were 22 mutual DEGs between the two datasets (Fig. 1C), which meant the DEGs observed in TCGA were also DEGs in GEO. Obviously from the heatmaps, COL1A1, COL10A1 and COL11A1 ranked in the top five among the up-regulated DEGs in both datasets (Figs. 1D–1I), further presented by boxplots. Likewise, COL4A4, COL6A5 and COL14A1 were the most down-regulated candidates (Figs. 1J–1O).

Figure 1 Differential expression analysis of collagen family genes between ESCC and normal tissues.

(A) and (B) Heatmaps of the DEGs in TCGA and GEO in descending order of logFC. The red and blue colors represent high and low expression, respectively. *P < 0.05; **P < 0.01; ***P < 0.001. (C) The Venn diagram showing the overlapped DEGs between the two datasets. (D–I) Boxplots of three representative up-regulated genes, COL1A1, COL10A1 and COL11A1 in TCGA and GEO. (J–O) Boxplots of three representative down-regulated genes, COL4A4, COL6A5 and COL14A1 in TCGA and GEO. DEG, differentially expressed gene; FC, fold change.

Survival analysis of collagen family genes in ESCC patients

HRs and P-values of the 22 DEGs were calculated and shown by heatmaps (Figs. 2A and 2B). Among them, HRs of COL6A5 and COL18A1 were the lowest and highest, respectively. Survival curves of the DEGs were plotted according to the Kaplan–Meier method. Consistently, COL6A5 and COL18A1 were the two genes most relevant to the overall survival of ESCC patients. Patients with lower COL6A5 expression exhibited poorer overall survival (P = 0.008 in TCGA, Fig. 2C; P = 0.060 in GEO, Fig. 2D). By contrast, patients with higher COL18A1 expression had worse overall survival (P = 0.393 in TCGA, Fig. 2E; P = 0.009 in GEO, Fig. 2F). These results suggested that COL6A5 and COL18A1 are tightly associated with the prognosis of ESCC.

Figure 2 Survival analysis of the DEGs of collagen family in ESCC patients.

(A) and (B) HRs and P-values of the DEGs related to overall survival in ascending order of HR in TCGA and GEO. (C) and (D) Kaplan–Meier survival curves of COL6A5 in TCGA and GEO. (E) and (F) Kaplan–Meier survival curves of COL18A1 in TCGA and GEO. DEG, differentially expressed gene; HR, hazard ratio.

DEGs-based prediction models to predict the prognosis of ESCC patients

Receiver operating characteristic curves have been extensively used to evaluate the predictive effect of one or more genes. The AUC value represents predictive accuracy and usually makes sense when it exceeds 0.60 (Lüdemann et al., 2006; Metz, 1978; Obuchowski, 2003). ROC curves of COL6A5 and COL18A1 indicated that good predictive performance could only be attained by COL6A5 in TCGA (AUC = 0.679, Fig. S1A), while COL18A1 had no predictive ability (Figs. S1C and S1D), suggesting that a single gene is not suitable for survival prediction of ESCC patients. Therefore, we established multi-gene prediction models based on expression levels of the DEGs to assess the joint effect of selected collagen genes on patient survival. There were seven genes in TCGA and nine genes in GEO finally included to form the models, respectively, and risk scores of the patients were calculated according to the below formulas: (2) Risk score(TCGA)=(1.528 * COL1A1Exp) + (0.265 * COL4A4Exp)     + (−0.539 * COL6A5Exp) + (−0.638 * COL11A1Exp)     +(−1.193 * COL12A1Exp) + (−0.244 * COL19A1Exp)     + (0.417 * COL24A1Exp)

(3) Risk score (GEO) = (7.700 * COL1A1Exp) + (-8.800 * COL1A2Exp)                                   + (-5.800 * COL3A1Exp) + (6.320 * COL5A1Exp)                                  + (-0.708 * COL6A5Exp) + (-0.790 * COL11A1Exp)                                  + (1.990 * COL14A1Exp) + (1.300 * COL22A1Exp)                                  + (2.400 * COL24A1Exp).

For instance, the positive coefficient for COL1A1 suggests that higher expression of COL1A1 was associated with worse survival. The negative value allocated to COL6A5 means that higher expression of COL6A5 was related to prolonged survival, in agreement with the survival analysis (Fig. 2). Notably, AUCs on the ROC curves of the DEGs-based models in TCGA and GEO reached 0.86 and 0.68, respectively (Figs. 3A and 3C), which were higher than those of the prediction models based on TNM staging in the two datasets with AUCs of 0.625 and 0.646, respectively (Figs. 3E and 3G). The TNM staging system is a generally recognized standard for classifying the spreading extent of cancer (D’Journo, 2018) and is commonly used to predict prognosis of cancer in clinical application. The prediction models, respectively, based on T-stage and N-stage were also examined but the AUCs were all less than 0.6 (Fig. S2). Furthermore, survival curves showed that patients with high risk were significantly correlated with poor survival (Figs. 3B, 3D, 3F and 3H). The 7-gene model in TCGA with true positive rate of 86% was more accurate than that of the TNM staging-based model, whereas predictive accuracy of the 9-gene model in GEO exhibited no difference. Therefore, the model in TCGA was used for our further studies. Finally, a heatmap was plotted to show the expression patterns of the seven genes in TCGA between high-risk and low-risk groups (Fig. 3I). The risk score distribution was exhibited in ascending order, and patients were divided into high- and low-risk groups by the median point (Fig. 3J). Overall, it can be seen that patients with high risk score had higher mortality rates and shorter survival time than those with low risk score (Fig. 3K). Taken together, above results indicated that the 7-gene model could be more accurate to predict patient survival.

Figure 3 Prediction models to predict the survival of ESCC patients.

(A–D) ROC and survival curves of the models based on expression of 7 and 9 collagen DEGs, respectively, in TCGA and GEO. (E–H) ROC and survival curves of the models according to TNM staging in TCGA and GEO. (I) A heatmap showing the expression patterns of the seven genes driving the prediction model in TCGA. (J) Risk score distribution of the patients in ascending order and divided into low-risk (green) and high-risk (red) in TCGA. (K) Survival time and status of the patients in order of increasing risk scores in TCGA. The red and green dots represent dead and alive, respectively. ROC, receiver operating characteristic; AUC, area under curve; DEG, differentially expressed gene; COL, collagen; TNM, tumor-node-metastasis.

Pathway analysis of collagen family genes

Gene sets enrichment analysis results showed that most of the gene sets were up-regulated in the high-risk group, and the top 20 enriched gene sets were given in Tables S3–S8. The gene sets that were closely associated with tumorigenesis were shown in Fig. 4. For instance, gene sets of PDGF, RB/P107, AKT/MTOR and p53 were significantly up-regulated according to Oncogenic Signatures Gene Sets (Figs. 4A–4F). Based on Hallmark Gene Sets, the enriched gene sets included p53 pathway, oxidative phosphorylation, apoptosis, mitotic spindle, G2/M checkpoint and notch signaling (Figs. 4G–4L). Using KEGG Gene Sets as reference, the high-risk group was tightly correlated with oxidative phosphorylation, renal cell carcinoma, bladder cancer, small cell lung cancer, adherens junction and cell cycle (Figs. 4M–4R).

Figure 4 GSEA results based on patient risk scores calculated by the prediction models in TCGA and GEO.

(A–F) Representative enriched gene sets according to Oncogenic Signatures Gene Sets. (G–L) Representative enriched gene sets according to Hallmark Gene Sets. (M–R) Representative enriched gene sets according to KEGG Gene Sets. GSEA, gene sets enrichment analysis. NES, normalized enrichment score.

Co-expression network analysis

WGCNA was performed to find out the genes that were co-expressed with collagen family genes in ESCC tissues. Risk-score-based DEGs that were differentially expressed between high- and low-risk groups were determined and presented by the volcano plot (Fig. S3). The co-expression network of collagen genes and the risk-score-based DEGs were given in several modules (Fig. 5). Collagen family genes were displayed as red nodes, and the genes included in the 7-gene prediction model in TCGA were marked as bigger red nodes. The blue nodes represented the co-expressed genes. Another network was drawn to show the association between collagen family genes and seven representative enriched gene sets (PDGF, RB/p107, PI3K/Akt/mTOR pathway, p53 pathway, oxidative phosphorylation, apoptosis and cell cycle) from the GSEA results (Fig. S4). The red nodes were the collagen family genes with close connections to those gene sets. A big blue circle represented a gene set and the blue nodes were genes included in each set. Genes closer to the center were more tightly associated with the collagen genes.

Figure 5 Co-expression network of collagen family genes.

Visualization of the co-expression between collagen family genes and the risk-scores-based DEGs. The red nodes are collagen family genes, and the bigger ones are the genes included in the 7-gene prediction model in TCGA. The blue nodes are the co-expressed genes. DEG, differentially expressed gene.

Discussion

Although extensive research efforts have been focused on this field in past decades, efficient detection methods for early ESCC and accurate prediction against complicated ESCC patients still remain an open issue. Recently, studies have found that the expression of certain genes, such as MCT4, ZNF750, Gli1, etc. was highly related to the occurrence and development of ESCC, and they might be applied as ideal biomarkers for ESCC (Cheng et al., 2018; Nambara et al., 2017; Yang et al., 2017; Zhang et al., 2018b). In addition, the aberrant expression of a few collagen family genes has also been reported to be significantly associated with the prognosis of ESCC patients. However, most works only focused on single or limited genes, and the predictive ability was barely satisfactory. Herein, we provided a more systematic analysis of the whole collagen family gene expression to evaluate the potential roles and clinical significance of collagen genes in ESCC.

We found that most of the collagen genes were up-regulated in ESCC tissues when compared to normal controls, half of which were identified as DEGs (Figs. 1A and 1B). Among them, the expression of COL1A1, COL10A1 and COL11A1 was particularly higher, and that of COL4A4, COL6A5 and COL14A1 was especially lower in tumor tissues, indicating their possible roles as diagnostic markers for ESCC. Consistently, several studies have shown that COL1A1, COL10A1 and COL11A1 were notably overexpressed in ESCC compared to normal tissues (Fang et al., 2019; He et al., 2017; Karagoz et al., 2016; Senthebane et al., 2018; Zhang et al., 2018a). Also, COL4A4 was also found to be down-regulated in esophageal tumor tissues (Chattopadhyay et al., 2009). Additionally, among the DEGs, COL7A1 was observed to be up-regulated in ESCC tissues (Kita et al., 2009). In our works, COL6A5, COL14A1 and some other collagen genes were reported to be significantly up- or down-regulated in ESCC tissues for the first time.

In the survival analysis, COL6A5 and COL18A1 were validated to be significantly related to overall survival of ESCC patients. Previous studies demonstrated that the COL6A5 expression was significantly associated with depressed behavior and atopic dermatitis (Söderhäll et al., 2007; Zhan et al., 2017), but no articles manifested its correlation with cancer. In addition, COL18A1 has been proved to be a promising biomarker for ovarian cancer and was possibly involved in the progression of bladder cancer (Fang et al., 2013; Peters et al., 2005). In this study, ESCC patients with low COL6A5 expression or high COL18A1 expression showed poor overall survival (Figs. 2C–2F), implying the expression of COL6A5 or COL18A1 as a potential indicator for the prognosis of ESCC patients. Moreover, the variations that affect the expression of COL6A5 and COL18A1 possibly have effects on the progression of ESCC. Activating COL6A5 or inhibiting COL18A1 might improve the therapeutic efficiency and the life-span of ESCC patients.

Because the expression of one gene is usually influenced by various factors, ideal effect may not be attained by using a single gene as a predictor. Indeed, COL6A5 achieved an AUC value over 0.60 only in TCGA (Fig. S1), making the requirement of another more powerful prediction method. Based on the selected collagen DEGs (7 genes in TCGA and 9 genes in GEO, both including COL6A5), we established two new prediction models. Importantly, such DEGs-based models exhibited better predictive ability than conventional prognostic models according to TNM staging. The 7-gene model in TCGA had especially higher predictive accuracy of 86%. One possible reason was that the RNA sequencing technology applied to TCGA was more accurate than the gene chip technology used in GEO. In summary, this 7-gene prediction model is greatly promising to predict the prognosis of ESCC patients and help determine next therapeutic regimens.

Furthermore, GSEA was used to identify significantly enriched gene sets and potentially relevant pathways (Fig. 4). The results showed that based on Oncogenic Signatures Gene Sets, gene sets of PDGF, RB/P107 and AKT/MTOR were significantly enriched in the high-risk group. It has been reported that PDGF receptor-beta increased the expression of COL1A2 through Akt/mTORC1 signaling pathway (Das et al., 2017). According to Oncogenic Signatures Gene Sets and Hallmark Gene Sets, the high-risk group was significantly related to p53 and p53 pathway, which suggested that collagen genes might be highly associated with the p53 or its related pathway in ESCC. Earlier studies proved that enhanced expression of ectopic p53 in dermal fibroblasts inhibited basal and TGF-beta-stimulated collagen gene expression, and the absence of cellular p53 was correlated with increased transcriptional activity of the Type I collagen gene (COL1A2) and collagen synthesis (Ghosh, Bhattacharyya & Varga, 2004). Moreover, the type IV collagen expression was inversely related to p53 in malignant tumors (Bar et al., 2004). Oxidative phosphorylation related genes were found to be up-regulated in the high-risk group by both Hallmark Gene Sets and KEGG Gene Sets. Indeed, some reports demonstrated that oxidative phosphorylation signature occurred when collagen density was decreased, and the change of collagen density microenvironment regulated the metabolism of cancer cells (Mah et al., 2018; Morris et al., 2016). As for apoptosis, an earlier study has shown that Type IV collagen could stimulate cancer cell proliferation, migration and inhibit apoptosis (Öhlund et al., 2013). Additionally, the gene sets of mitotic spindle, G2/M checkpoint and cell cycle were enriched in the high-risk group as well, implying that collagen might regulate the cell cycle of ESCC cells. Furthermore, it was indicated that the high-risk group was markedly associated with renal cell carcinoma, bladder cancer and small cell lung cancer. These results were consistent with previous studies that collagen gene expression was correlated with the poor prognosis of those cancers (Koskimaki et al., 2010; Wan et al., 2015; Xu et al., 2017; Zeng et al., 2018).

As shown by the co-expression network (Fig. 5), a few collagen family genes such as COL1A1, COL11A1, COL6A6 and COL19A1, were co-expressed with NETO1, NEUROD2 and NRG3, which are the genes involved in neural functions. These findings could be verified by earlier articles to some extent (McCarthy & Hay, 1991; Perris et al., 1993a, 1993b). COL11A1 was also observed to be co-expressed with tumor suppressor candidate 7 (TUSC7), further validating the possible role of COL11A1 in the occurrence of ESCC. Beyond that, some potassium channel related genes (KCNA2, KCNE1B, KCNH1, KCNJ4 and KCNK4) were co-expressed with collagen genes in a way, revealing that collagen genes might be correlated with the regulation of potassium channels in ESCC. As for the two potential prognostic biomarkers, COL18A1 only showed close relations with collagen family members, while COL6A5 was associated with two other genes in this network, ROBO2 and MIR548A3. ROBO2 has been identified as a candidate tumor suppressor (Trifonov et al., 2013), and the alteration of its expression might play a role in malignant tumors of digestive tract including gastric and colorectal cancers (Je et al., 2013).

Apart from what is aforementioned, there are still some limitations of this research. For instance, the prediction model was comprised of several genes, making it difficult to conduct cellular experiments by targeting a single gene to confirm its predictive effect. Aside from it, the characteristics of patient samples, as well as the methodology utilized in TCGA, were somewhat different from that in GEO, which may explain the different results coming from the two datasets. For example, TCGA uses the RNA sequence technology while GEO applies the gene chip technology to detect gene expression of patient tissues. Besides, TCGA mainly collected data from white people, whereas the majority of patients in GEO (GSE53625) were Asian. Therefore, there was no a single gene that exhibited significant P-values in both datasets in the survival analysis, and the selected genes driving the prediction model in one dataset were not completely identical to those in another dataset. Further validation of these outcomes requires more clinical information and biological experiments in the future.

Conclusions

In summary, this study identified 22 collagen family genes that were significantly expressed higher or lower in ESCC compared to normal tissues. Among them, COL1A1, COL10A1, COL11A1, COL4A4, COL6A5 and COL14A1 were the most distinct ones and possessed the potential in ESCC diagnosis. Besides, COL6A5 and COL18A1 showed strong correlations with overall survival of ESCC patients and might be robust prognostic biomarkers for ESCC. Furthermore, we established a 7-gene prediction model with high performance to predict the prognosis of ESCC patients. In terms of the underlying mechanism, collagen genes might be associated with PI3K/Akt/mTOR pathway, p53 pathway, oxidative phosphorylation, apoptosis and cell cycle during the progression of ESCC. Our works may further benefit the diagnosis, prognosis and treatments for ESCC patients.

Supplemental Information

Supplemental Information 1 The results of differential expression analysis of collagen family genes in TCGA.

A total of 44 collagen family genes are listed in descending order of logFC. Genes with P < 0.05 and FDR < 0.05 were considered as differentially expressed genes (DEGs). logFC, log2 (fold change); logCPM, log2 (counts per million); FDR, false discovery rate.

Click here for additional data file.

Supplemental Information 2 The results of differential expression analysis of collagen family genes in GEO.

A total of 44 collagen family genes are listed in descending order of logFC. Genes with P < 0.05 and FDR < 0.05 were considered as differentially expressed genes (DEGs). logFC, log2 (fold change); AveExpr, Average expression; t, t-statistic; adj.P.Val, adjusted P-value; B, B-statistic.

Click here for additional data file.

Supplemental Information 3 The top 20 enriched gene sets according to Oncogenic Signatures Gene Sets in TCGA.

Oncogenic Signatures Gene sets are signatures of cellular pathways usually dis-regulated in cancer. The top 20 gene sets are listed in descending order of NES, which reflects the degree to which a gene set was overrepresented in the groups. Gene sets with P < 0.05 and FDR < 0.25 were considered to be significant. ES, enrichment score; NES, normalized enrichment score; NOM p-val, nominal P-value; FDR, false discovery rate.

Click here for additional data file.

Supplemental Information 4 The top 20 enriched gene sets according to Oncogenic Signatures Gene Sets in GEO.

Oncogenic Signatures Gene sets are signatures of cellular pathways usually dis-regulated in cancer. The top 20 gene sets are listed in descending order of NES, which reflects the degree to which a gene set was overrepresented in the groups. Gene sets with P < 0.05 and FDR < 0.25 were considered to be significant. ES, enrichment score; NES, normalized enrichment score; NOM p-val, nominal P-value; FDR, false discovery rate.

Click here for additional data file.

Supplemental Information 5 The top 20 enriched gene sets according to Hallmark Gene Sets in TCGA.

Hallmark Gene Sets summarize and represent specific well-defined biological states or processes and display coherent expression. The top 20 gene sets are listed in descending order of NES, which reflects the degree to which a gene set was overrepresented in the groups. Gene sets with P < 0.05 and FDR < 0.25 were considered to be significant. ES, enrichment score; NES, normalized enrichment score; NOM p-val, nominal P-value; FDR, false discovery rate.

Click here for additional data file.

Supplemental Information 6 The top 20 enriched gene sets according to Hallmark Gene Sets in GEO.

Hallmark Gene Sets summarize and represent specific well-defined biological states or processes and display coherent expression. The top 20 gene sets are listed in descending order of NES, which reflects the degree to which a gene set was overrepresented in the groups. Gene sets with P < 0.05 and FDR < 0.25 were considered to be significant. ES, enrichment score; NES, normalized enrichment score; NOM p-val, nominal P-value; FDR, false discovery rate.

Click here for additional data file.

Supplemental Information 7 The top 20 enriched gene sets according to KEGG Curated Gene Sets in TCGA.

KEGG Gene Sets are derived from the KEGG pathway database. The top 20 gene sets are listed in descending order of NES, which reflects the degree to which a gene set was overrepresented in the groups. Gene sets with P < 0.05 and FDR < 0.25 were considered to be significant. ES, enrichment score; NES, normalized enrichment score; NOM p-val, nominal P-value; FDR, false discovery rate.

Click here for additional data file.

Supplemental Information 8 The top 20 enriched gene sets according to KEGG Curated Gene Sets in GEO.

KEGG Gene Sets are derived from the KEGG pathway database. The top 20 gene sets are listed in descending order of NES, which reflects the degree to which a gene set was overrepresented in the groups. Gene sets with P < 0.05 and FDR < 0.25 were considered to be significant. ES, enrichment score; NES, normalized enrichment score; NOM p-val, nominal P-value; FDR, false discovery rate.

Click here for additional data file.

Supplemental Information 9 Single-gene prediction models based on the gene expression of COL6A5 and COL18A1 in TCGA and GEO datasets.

(A) and (B) ROC curves of COL6A5 in the two datasets. (C) and (D) ROC curves of COL18A1 in the two datasets.

Click here for additional data file.

Supplemental Information 10 Prediction models according to T-stage and N-stage of the patients in TCGA and GEO datasets.

(A) and (B) ROC curves of T-stage in the two datasets. (C) and (D) ROC curves of N-stage in the two datasets.

Click here for additional data file.

Supplemental Information 11 Volcano plot showing the DEGs that were differentially expressed between high- and low-risk groups based on the risk scores calculated by the 7-gene prediction model in TCGA.

Red and green dots indicate up- and down-regulated genes, respectively.

Click here for additional data file.

Supplemental Information 12 Co-expression network of collagen genes and seven significantly enriched gene sets obtained by GSEA.

Red nodes are the collagen family genes closely correlated with those gene sets. A big blue circle represented a gene set and the blue nodes were genes included in each gene set.

Click here for additional data file.

Supplemental Information 13 Gene expression profiles of tumor and normal samples in ESCC patients from the TCGA database.

Click here for additional data file.

Supplemental Information 14 Clinical data of ESCC patients from the TCGA database.

Click here for additional data file.

Additional Information and Declarations

Competing Interests

Author Contributions

Data Availability

The authors declare that they have no competing interests.

Jieling Li performed the experiments, analyzed the data, contributed reagents/materials/analysis tools, prepared figures and/or tables, authored or reviewed drafts of the paper, approved the final draft.

Xiao Wang performed the experiments, analyzed the data, prepared figures and/or tables.

Kai Zheng authored or reviewed drafts of the paper.

Ying Liu prepared figures and/or tables.

Junjun Li contributed reagents/materials/analysis tools.

Shaoqi Wang collected data.

Kaisheng Liu gave advice on this study.

Xun Song analyzed the data, authored or reviewed drafts of the paper, approved the final draft, gave advice on the revision of manuscript.

Nan Li collected data.

Shouxia Xie conceived and designed the experiments, authored or reviewed drafts of the paper, approved the final draft.

Shaoxiang Wang conceived and designed the experiments, contributed reagents/materials/analysis tools, prepared figures and/or tables, authored or reviewed drafts of the paper, approved the final draft.

The following information was supplied regarding data availability:

Raw data was downloaded from public databases including TCGA and GEO (GSE53625). The raw data from TCGA is available as Supplemental Files.

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
