# Peer review of "The clinical significance of collagen family gene expression in esophageal squamous cell carcinoma"

_PeerJ, doi:10.7717/peerj.7705_

## Round 0.1 · original submission · Major Revisions

Please go through the comments of the reviewers and make the requested changes, all of which will help this manuscript.

[]

Reviewer 1 ·

Basic reporting

Authors analyze clinical data and gene expression profiles of Esophageal squamous cell carcinoma (ESCC) patients from two databases, The Cancer Genome Atlas (TCGA) and Gene Expression Omnibus (GEO) to look for relevant collagen genes as potential biomarkers for the disease. The overall language used in the manuscript is satisfactory with minor rephrasing required in few sentences. The introduction provides a brief background however it should be expanded mostly on the need to look for biomarkers of ESCC and if these markers may be helpful in early detection. The discussion needs more expansion in a couple of directions, mostly related to results obtained from pathway analysis and co-expression network analysis. Also, throughout the manuscript authors are not elaborative with the specific terms used, these terms should be defined.

Experimental design

Based on the Clinicopathological information of the ESCC patients seems like, TNM and T-stage information is significantly correlated with survival in both databases, TCGA and GEO. Why have authors left out T-stage prediction model? The model should also be looked for T-stage, as is for TNM staging based model for both the databases.

Validity of the findings

Authors should elaborate on the basis for the selection of final 7 genes from the TCGA and 9 genes from the GEO and clarify in the text. Were individual AUC's for all genes selected for DEG based forecast model is above 0.6. Also, it’s not clear how the threshold was set.

In the TCGA database, there were 81 tumor samples and 11 normal samples. Authors should justify that such a difference between no’s will not put a bias in the study.

Additional comments

Major comments
Based on the Clinicopathological information of the ESCC patients seems like, TNM and T-stage information is significantly correlated with survival in both databases, TCGA and GEO. Why have authors left out T-stage prediction model? The model should also be looked for T-stage, as is for TNM staging based model for both the databases.

Please elaborate on the basis for the selection of final 7 genes from the TCGA and 9 genes from the GEO and clarify in the text. Was individual AUC for all genes selected for DEG based forecast model is above 0.6. Also, it’s not clear how the threshold was set.

Significantly enriched gene sets were obtained according to Oncogenic Signatures Gene Sets, Hallmark Gene Sets and KEGG Gene Sets. Authors should put information on NES, and how it is interpreted. Overall there doesn’t seem to be much consensus between 3 different gene sets, except for some overlap in p53 and oxidative phosphorylation. Authors touch the base here or some elaboration on possible discussion role of oxidative phosphorylation. However, authors should put forward a discussion on how functionally and mechanistically p53 possibly correlates to collagen gene expression in tumors. As there is evidence that p53 is implicated in connective tissue homeostasis.

Authors should clearly define and provide a brief background on some of the terms used in this manuscript, such as Hazard ration, GSEA, WCGNA, T-stage, N-stage, NES, etc, to make it clear to readers.

In TCGA database there were 81 tumor samples and 11 normal samples. Authors should justify that such a difference between no’s will not put a bias in the study.

For co-expression network analysis by WCGNA, authors should elaborate in discussion the findings from fig 5. Such as what key genes seemed to be co-expressing with low risk and high-risk gene sets. Was there any pattern observed?

Minor points

Lines, 53-54, “Thus, it is urgent to identify novel biomarkers and therapeutic targets to facilitate the development of treatments”. Biomarkers will also be helpful in early detection. Also, authors should shed some light on what’s the prognosis and chances of survival if these tumors are detected at an early stage.

Line 68, correct typo “patway”

Line 89-90, “All the gene expression levels 90 were normalized before studying”. Normalized to what?

Lines 98-100, sentence not clear

Reviewer 2 ·

Basic reporting

No Comment

Experimental design

No Comment

Validity of the findings

No Comment

Additional comments

In this work to investigate the significance of collagen genes as potential biomarkers for the diagnosis and prognosis of Esophageal squamous cell carcinoma (ESCC), the authors carried out gene expression analysis of the 44 members of the collagen gene family in two datasets obtained from TCGA and GEO. They propose a DEG-based forecast model for predicting the survival of ESCC patients based on 7 collagen genes.

Line 89-90: All the gene expression levels were normalized before studying – How? Give details.

Line 92: using which statistical methods?

Line 101: Genes most related to overall survival of patients were shown?? Figure no?

Line 104: Multivariate Cox regression model was used to establish forecast models based on gene expression of the DEGs and overall survival of the patients via R software - which R software?

Line 105: Formulas and risk score of each patient were obtained? What formulas – not clear.

Line no. 112-113: Gene sets enrichment analysis (GSEA) was performed to explore potential mechanism of collagen family genes by dividing the patient samples into high-risk and low-risk groups. Not clear. As I understand, 22 collagen genes were submitted to GSEA portal and important pathways identified.

Line no. 115: Some software javaGSEA is used. However no information is given about what is does, how it helps in identifying gene sets enriched in low and high risk groups, what are these gene sets (no legends provided for the supplementary tables). Not clear, please provide details.

Line no. 121-123: Risk-score-based DEGs were determined by dividing the patient samples into high-risk and low risk groups in TCGA and WGCNA performed to assess the relationships between collagen family genes and the risk-score-based DEGs as well as seven representatively enriched gene sets – what are these 7 enriched gene sets? How are risk-score based DEGS identified - not clear. Are these only collagen genes, or other genes as well – not clear. There are two sets of risk-scores, one based on 7 collagen genes for TCGA dataset, and another similar for GEO dataset – this is used to divide the patients in each dataset into low and high risk groups – right?

Line no. 139: what statistical methods – explicitly mention what analysis was done.

Line no. 166: How were the 7 genes in TCGA dataset and 9 genes in GEO dataset selected – specify. Any common genes between the two datasets? Why is this dataset dependent – discuss.

Line no. 168: In the formula of Risk scores, what are the multipliers (coefficients multiplied with the gene expression value of collagen genes) – please specify. Also give some justification of defining the risk score.

Line no. 184-185: No difference in the expression profile of the 7-genes in the low-risk and high-risk groups observed. Discuss.

Line no. 187-189: Fig 3G is not very conclusive, even patients with low risk score have low survival rates as for high risk score patients.

Line no. 192: Not clear what genes were submitted to GSEA? All Collagen genes? Only those collagen genes that were significant DEGs in high-risk group? Please provide details The whole GSEA analysis is very confusing. Provide details of how analysis was performed.

In the WGCNA analysis, how are the COL genes distributed across the modules? Discuss the functional enrichment analysis of genes co-expressed with COL genes from these modules. How the Fig S4 is obtained not clear. Provide details.

There is confusion in the analysis of the two datasets presented, one from TCGA and the other from GEO. It is not clear whether the DEGs observed in TCGA dataset are also DEGs in GEO dataset. Please clarify. Further, it is mentioned that TCGA data being based on RNASeq compared to microarray-based GEO data, the 7-gene forecast model is more accurate. How does it perform on the GEO dataset? That would provide a good validation of the proposed model. The difference in the ethnicity of the patients is being discussed as possible reason for no overlap between the two datasets. In that case it would have been more appropriate to consider datasets with similarity in the ethnic group of the patients considered.

In general, details of all the analyses performed are missing.

No Table legends provided for the supplementary tables. Similarly for supplementary figures, no legends provided.

English of the manuscript needs to be improved before publication. Also run a spellcheck before resubmission.

Significant amount of analysis is done but since no details are provided how it is done, it cannot be accepted in the current form.

Reviewer 3 ·

Basic reporting

The objective of the study is good as it focuses on the “The identification of novel biomarkers for early identification of the disease to facilitate the effective treatment for esophageal squamous cell carcinoma.” However, the manuscript has to be revised thoroughly so as to reach the standards of the journal.

Experimental design

Regarding the datasets used for the work, more details regarding the controls used and number of replicates used should be provided.

Validity of the findings

No comment

Additional comments

All over the manuscript, there are grammatical errors and the manuscript should be subjected to English language correction by a native speaker.

---

## Round 0.2 · Minor Revisions

I request you to make the remaining minor changes recommended by reviewers.

Reviewer 1 ·

Basic reporting

Satisfied with basic reporting

Experimental design

Scientifically sound

Validity of the findings

Findings well described with proper rationale.

Additional comments

Authors have revised the manuscript properly and have answered all queries raised.
I recommend accepting the manuscript.

Reviewer 2 ·

Basic reporting

OK

Experimental design

OK

Validity of the findings

OK

Additional comments

The authors have revised the paper and most of the comments have been addressed. However, the following two comments are still remaining.

1) Line 120: Prediction models
The justification for the use of risk-scores is still missing. Probably, a reference may be cited in the manuscript.

2) Line 229: Pathway analysis of Collagen genes
In reply20/21, the authors have mentioned that they have categorized samples into 'High' and 'Low' risk groups and then carried out GSEA across all the genes (not just collagen genes), while the title is misleading. Again, in the discussion, they focussed on explaining collagen genes and their involvement in pathways of cancer. For clarity, the authors may add what set of genes were used for gene enrichment analysis using GSEA, as the current heading indicates only collagen genes were submitted for enrichment analysis. Some details is required.

---

## Round 0.3 · accepted · Accept

Good News, this article is accepted!

#